elLIFE

# The genetic architecture of gene expression levels in wild baboons

Jenny Tung[1]*[†abc], Xiang Zhou[1,2†d], Susan C Alberts[3,4], Matthew Stephens[1,2], Yoav Gilad[1]*

[1]Department of Human Genetics, University of Chicago, Chicago, United States; [2]Department of Statistics, University of Chicago, Chicago, United States; [3]Institute of Primate Research, National Museums of Kenya, Nairobi, Kenya; [4]Department of Biology, Duke University, Durham, United States

**Abstract** Primate evolution has been argued to result, in part, from changes in how genes are regulated. However, we still know little about gene regulation in natural primate populations. We conducted an RNA sequencing (RNA-seq)-based study of baboons from an intensively studied wild population. We performed complementary expression quantitative trait locus (eQTL) mapping and allele-specific expression analyses, discovering substantial evidence for, and surprising power to detect, genetic effects on gene expression levels in the baboons. eQTL were most likely to be identified for lineage-specific, rapidly evolving genes; interestingly, genes with eQTL significantly overlapped between baboons and a comparable human eQTL data set. Our results suggest that genes vary in their tolerance of genetic perturbation, and that this property may be conserved across species. Further, they establish the feasibility of eQTL mapping using RNA-seq data alone, and represent an important step towards understanding the genetic architecture of gene expression in primates.

*For correspondence: jt5@duke.edu (JT); gilad@uchicago.edu (YG)

[†]These authors contributed equally to this work

Present address: [a]Department of Evolutionary Anthropology, Duke University, Durham, United States; [b]Duke Population Research Institute, Duke University, Durham, United States; [c]Institute of Primate Research, National Museums of Kenya, Nairobi, Kenya; [d]Department of Biostatistics, University of Michigan, Ann Arbor, United States

Competing interests: The authors declare that no competing interests exist.

## Introduction

Gene regulatory variation has been shown to make fundamental contributions to phenotypic variation in every species examined to date. This relationship has been demonstrated most clearly at the level of gene expression, which captures the integrated output of a large suite of other regulatory mechanisms. Variation in gene expression levels has been linked to fitness-related morphological, physiological, and behavioral variation in both lab settings and natural populations (e.g., *Abzhanov et al., 2004*; *Hammock and Young, 2005*; *Tishkoff et al., 2006*; *Chan et al., 2010*; reviewed in *Wray, 2007*), and is a robust biomarker of disease in humans (e.g., *Golub et al., 1999*; *Borovecki et al., 2005*). In addition, patterns of gene expression are often associated with signatures of natural selection (*Rifkin et al., 2003*; *Denver et al., 2005*; *Gilad et al., 2006a*; *Blekhman et al., 2008*), suggesting their functional importance even when their phenotypic significance remains unknown.

In primates, the majority of research on the evolution of gene expression has concentrated on cross species comparisons, particularly using humans, chimpanzees, and rhesus macaques (*Enard et al., 2002*; *Cáceres et al., 2003*; *Khaitovich et al., 2004*; *Gilad et al., 2005, 2006b*; *Haygood et al., 2007*; *Blekhman et al., 2008*; *Babbitt et al., 2010*; *Barreiro et al., 2010*; *Blekhman et al., 2010*; *Brawand et al., 2011*; *Perry et al., 2012*). These studies—motivated by a long-standing argument about the importance of gene regulation in primate evolution (*King and Wilson, 1975*)—have been important for identifying patterns of constraint on gene expression phenotypes over long evolutionary time scales, and for suggesting candidate loci that might contribute to phenotypic uniqueness in humans or other species. For example, gene expression patterns associated with neurological development appear to have experienced an accelerated rate of change in primates relative to other

**eLife digest** Our genes contain the instructions needed to make all aspects of the body. These instructions can be changed by altering the sequence of the DNA that makes up the genes, which can account for many of the different characteristics found in humans and other animals.

However, our characteristics can also be altered by changing how often the genes issue their instructions, which is known as gene expression. For example, it is thought that changes in the expression of some genes in primates may account for the expansion of brain sizes over evolutionary time, particularly in the ancestors of modern humans. Most studies into gene expression in primates have compared different species or focused on humans. It is less clear how many, and what type of, genes vary in expression between individuals of the same species in other natural populations.

Here, Tung, Zhou et al. used a technique called RNA-sequencing to study gene expression in a population of wild baboons that have been studied for over four decades by the Amboseli Baboon Research Project in Kenya. This involved collecting blood samples from 63 individually recognized adult baboons. After RNA-sequencing, Tung, Zhou et al. were able to identify specific sections of the baboon genome where the DNA sequence an individual baboon carried could predict how highly individual genes were expressed. These sections are known as 'expression quantitative trait loci' (or eQTLs for short).

Tung, Zhou et al. found that there was a lot of genetically controlled variation in gene expression across the 63 baboons. Most of the eQTLs were found to be in genes that are rapidly evolving or are relatively new. There were fewer eQTLs in genes that are shared across a wide variety of species, possibly because keeping the expression of these genes stable is important for processes that are essential for life.

Many of the eQTLs found in the baboons were in genes where eQTLs are also found in humans. This suggests that the set of genes where genetic variation affects gene expression in the baboons may also be a similar set in humans.

Tung, Zhou et al. also examined how the age, sex, and social integration of the baboons affected the variation in gene expression observed in the population. They found that for most genes, these factors had only small effects on gene expression levels. However, for some genes, these factors could affect the level of expression throughout the life of the individual.

These findings demonstrate that it is feasible to study gene expression patterns in wild primates. The next challenge is to investigate how environmental and genetic factors combine to influence gene expression, and the evolutionary impact of these effects for animals as a whole.

mammals, with axonogenesis-related and cell adhesion-related genes accelerated specifically in the human lineage (*Brawand et al., 2011*). Similarly, differentially expressed genes in human liver are enriched for metabolic function (*Blekhman et al., 2008*), suggesting a potential molecular basis for arguments implicating dietary shifts in the emergence of modern humans (*Kaplan et al., 2000*; *Ungar and Teaford, 2002*; *Wrangham, 2009*).

Adaptively relevant changes in gene expression levels across species implicate selection on gene expression phenotypes within species, and particularly within populations, the basic unit of evolutionary change. However, in contrast to cross species comparisons, we still know little about the genetic architecture of gene expression levels in natural nonhuman primate populations. No estimates of the heritability of gene expression traits are available, even for populations that have been intensively studied for many decades. We also do not know whether segregating genetic variation that affects gene expression is common or rare, how the effect sizes of such variants are distributed, or whether they carry a signature indicative of natural selection. If gene regulatory variation has indeed been key to primate evolution, as classic arguments suggest (*King and Wilson, 1975*), then large gaps therefore remain in our understanding of this process.

Three primary reasons combine to account for the absence of such data. First, until relatively recently, the only feasible approach for measuring genome-wide gene expression levels on a population scale was microarray technology. This constraint limited the diversity of systems that could

be assessed because cost-effective, commercially available arrays have only been developed for a handful of taxa. Second, genomic resources, especially detailed catalogs of known genetic variants (e.g., *1000 Genomes Project Consortium et al., 2010*, *International HapMap Consortium, 2005*), are also limited to a small set of species. The lack of such resources creates major barriers to genome-scale studies of the genetics of gene expression in other organisms, which rely on complementary gene expression and genotype data. Finally, for many taxa, samples suitable for gene expression profiling can be challenging to collect. In nonhuman primates, for example, RNA samples are rarely available even for the most intensively studied natural populations.

Recently, sequencing-based methods for measuring gene expression levels (e.g., RNA-seq) have eliminated the need for species-specific arrays. Comparative genomic studies using RNA-seq have thus vastly expanded the set of taxa for which genome-wide expression data are available (including primates: *Brawand et al., 2011*; *Perry et al., 2012*). Importantly, because fragments of expressed genes are resequenced many times in RNA-seq studies, data on genetic variation are also generated in the process. Although these data can be affected by technical biases, several studies have demonstrated the generally high reliability of genotypes inferred from RNA-seq reads (*Perry et al., 2012*; *Piskol et al., 2013*). Such data can provide important insight into genetic diversity in species for which little other information exists (*Perry et al., 2012*). Additionally, they provide the two ingredients necessary for mapping gene expression traits to genotype, at moderate cost and without the requirement for previously ascertained genetic variants.

Here, we evaluate the potential for such work in an intensively studied wild primate population, the baboons (*Papio cynocephalus*) of the Amboseli basin in Kenya. 43 years of prior research on this population have established it as an important model for human social behavior, health, and aging (*Alberts and Altmann, 2012*), and have facilitated the development of protocols for collecting samples appropriate for gene expression analysis (*Tung et al., 2009*; *Babbitt et al., 2012*; *Runcie et al., 2013*). We generated RNA-seq data for 63 individually recognized members of the Amboseli study population. We used these data to explore the frequency, impact, and potential selective relevance of variants associated with variation in gene expression levels, using complementary expression quantitative trait locus (eQTL) mapping and allele-specific expression (ASE) approaches. We found evidence for abundant functional regulatory variation in the Amboseli baboons, and a surprising amount of power to detect these variants even with a modest sample size. We also found that functional variants are depleted in highly conserved genes, consistent with constraint on gene expression patterns. However, among genes with eQTL, we did not find strong support for a relationship between effect size and minor allele frequency. Such a relationship would be consistent with pervasive negative selection on gene expression phenotypes (i.e., selection against variants that produce large perturbations in gene expression levels) and has been suggested by work in humans (*Battle et al., 2014*). Finally, we used our data set to provide the first estimates of the heritability of gene expression levels in wild primates, including the relative contributions of *cis*-acting and *trans*-acting genetic variation.

## Results

### Functional regulatory variation is common in the Amboseli baboons

We obtained blood samples from 63 individually recognized adult baboons in the Amboseli population (*Figure 1—figure supplement 1*). From these samples, we produced a total of 1.89 billion RNA-seq reads (mean of 30.0 ± 4.5 s.d. million reads per individual, with 8.6 ± 1.8 s.d. million reads uniquely mapped to exons: *Supplementary file 1A*). On average, 67.2% of reads mapped to the most recent release of the baboon genome (*Panu2.0*), 69.2% of which could be assigned to a unique location. We used the set of uniquely mapped reads to estimate gene-wise gene expression levels for NCBI-annotated baboon RefSeq genes. After subsequent read processing and normalization steps ('Materials and methods', *Figure 1—figure supplement 1* and *Figure 1—figure supplement 2*), we considered variation in gene expression levels for 10,409 genes expressed in whole blood (i.e., all genes for which we could test for *cis*-acting genetic effects on gene expression).

We also used the RNA-seq reads to identify segregating genetic variants in the Amboseli population. We considered only high confidence sites that were variable within the Amboseli population ('Materials and methods'; *Figure 1—figure supplement 3*). As expected (*Piskol et al., 2013*), these sites were highly enriched in annotated gene bodies (*Figure 1*; *Figure 1—figure supplement 4*).

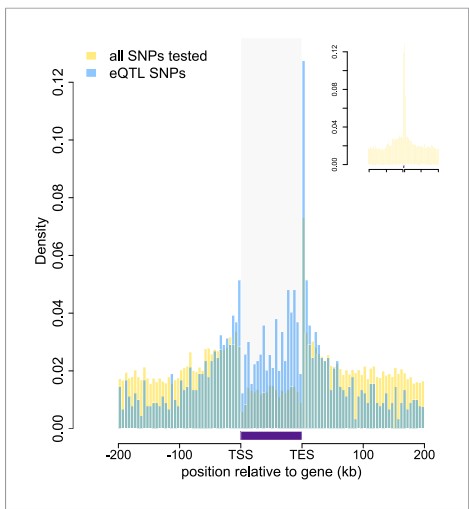

**Figure 1**. Baboon eQTLs are enriched in and near genes. The locations of all SNPs tested in the eQTL analysis are shown in gold relative to the 5′ most gene transcription start site (TSS) and the 3′ most gene transcription end site (TES) for all 10,409 genes. SNPs detected as eQTL are overplotted in blue, and are enriched, relative to all SNPs tested, near transcription start sites, transcription end sites, and within gene bodies. Gray shaded rectangle denotes the region bounded by the TSS and TES, with gene lengths divided into 20 bins for visibility (because the gene body is thus artificially enlarged, SNP density within genes cannot be directly compared with SNP density outside of genes). Note that SNPs that fall outside of one focal gene may fall within the boundaries of other genes. Inset: distribution of all SNPs tested relative to the location of genes, highlighting the concentration of SNPs in genes (the peak at the center of the plot). See *Figure 1—figure supplements 1–14* for additional details on workflow, variant calling validation, location of all analyzed SNPs relative to genes, agreement between eQTL and ASE detection, and effects of local structure.

The following figure supplements are available for figure 1:

**Figure supplement 1**. Detailed workflow for gene expression level estimation.

**Figure supplement 2**. Elimination of GC bias via quantile normalization.

**Figure supplement 3**. Detailed workflow for SNP genotyping.

**Figure supplement 4**. Location of analyzed SNPs relative to genes.

**Figure supplement 5**. Accuracy of genotype calls for SNPs independently typed in HapMap3.

Based on parallel analyses applied to human RNA-seq data, we estimated approximately 97% of these sites to be true positives, and a median correlation between true genotypes and inferred genotypes of 98.7% ('Materials and methods'; *Figure 1—figure supplements 5–6*). To identify putative expression quantitative trait loci (eQTL), we focused on variants that passed quality control filters, within 200 kb of the gene of interest. Such variants represent likely *cis*-acting eQTL, which are more readily identifiable in small sample sizes than *trans*-eQTL. To identify cases of allele-specific expression, which provides independent but complementary evidence for functional *cis*-regulatory variation, we focused on genes for which multiple heterozygotes were identified for variants in the exonic regions of expressed genes. We also required a minimum total read depth at exonic heterozygous sites of 300 reads (which should provide high power to detect modest ASE: *Fontanillas et al., 2010*), resulting in a total set of 2280 genes tested for ASE.

Both analyses converged to reveal extensive segregating genetic variation affecting gene expression levels in the Amboseli population. At a 10% false discovery rate, we identified eQTL for 1787 (17.2%) of the genes we analyzed, and evidence for ASE for 510 (23.4%) of tested genes. Consistent with reports in humans (e.g., *Veyrieras et al., 2008*; *Pickrell et al., 2010a*), eQTL were strongly enriched near gene transcription start sites and in gene bodies (*Figure 1*; controlling for the background distribution of sites tested, which were also enriched in and around genes). Within gene bodies, eQTL were particularly likely to be detected near transcription end sites; this potentially reflects enrichment in 3′ untranslated regions, which are poorly annotated in baboon. Also as expected, genes with eQTL were more likely to exhibit significant ASE and vice-versa (hypergeometric test: $p < 10^{-25}$; *Figure 1—figure supplement 7*). The magnitude and direction of ASE and eQTL were significantly correlated when an eQTL SNP could also be assessed for ASE (n = 123 genes; $r = 0.719$, $p < 10^{-20}$, *Figure 1—figure supplement 7*), and when ASE SNPs were assessed as eQTL (n = 510 genes; $r = 0.575$, $p < 10^{-45}$, *Figure 1—figure supplement 7*). Detection of ASE was most strongly favored for highly expressed genes (i.e., higher RPKM: Wilcoxon test: $p < 10^{-208}$; *Figure 1—figure supplement 8*), whereas detection of eQTL was most strongly favored for genes with high

*Figure 1. Continued*

**Figure supplement 6**. PCA projection of YRI samples using the RNA-seq-based pipeline vs independently typed SNPs.

**Figure supplement 7**. Agreement between eQTL and ASE approaches for identifying functional variants.

**Figure supplement 8**. Power to detect ASE vs eQTL.

**Figure supplement 9**. Characteristics of YRI eQTL identified in the RNA-seq vs conventional pipelines.

**Figure supplement 10**. Differences in the magnitude of ASE vs distance between sites.

**Figure supplement 11**. Location of eQTL SNPs relative to genes with and without controlling for local structure.

**Figure supplement 12**. Number of eQTL identified by PCs removed from the gene expression data set.

**Figure supplement 13**. Coverage by genotype call.

**Figure supplement 14**. Detection of ASE is not dependent on number of heterozygotes, conditional on total read depth.

local SNP density (p < $10^{-72}$; *Figure 1—figure supplement 8*).

## Increased power to detect eQTL in baboons relative to humans

The number and effect sizes of the eQTL we detected indicate that our power to detect eQTL in the Amboseli population was surprisingly high, especially given that our genotyping data set was limited only to those sites represented in RNA-seq data (i.e., primarily within transcribed regions of moderately to highly expressed genes). Further, while thousands of *cis*-eQTL have been mapped in single human populations, doing so has generally required sample sizes several fold larger than ours (*Lappalainen et al., 2013*; *Battle et al., 2014*).

To provide a more informative estimate of the difference in power to detect eQTL in baboons relative to humans, we applied the same mapping, data processing, variant calling, and eQTL modeling pipeline to a similarly sized RNA-seq data set on 69 Yoruba (YRI) HapMap samples, in which samples were sequenced to a similar depth (*Pickrell et al., 2010a*). Using our approach for estimating and modeling the gene expression data, but obtaining the genotype data from an independent array platform, we could identify 700 genes with significant eQTL in the YRI data set at a 10% FDR. Approximately half (51%) could be recovered if we only focused on SNPs in transcribed regions. This number (n = 357) therefore reflects the likely theoretical limit of detection for performing eQTL mapping in which SNPs are called based on RNA-seq data. Indeed, when eQTL mapping for the YRI was conducted using genotype data obtained from RNA-seq reads (i.e., the same pipeline used for the baboons), we identified 290 genes with eQTL (41.4% of those identified using independently collected genotype data). eQTL identified in the RNA-seq pipeline do not differ from those identified only in the conventional pipeline in either effect size or in surrounding sequence conservation, but do tend to fall in more highly expressed genes (Wilcoxon test on RPKM values: p = 6.53 × $10^{-9}$; *Figure 1—figure supplement 9*), suggesting that sequencing coverage considerations reduce the number of identifiable eQTL below the theoretical maximum. The RNA-seq-based pipeline therefore reduces the number of genes with detectable eQTL by 50–60%, suggesting that if genotyping array data had been available for the baboons, we might have identified eQTL for ~3500–4000 genes, comparable to results from human data sets with more than 350 samples (*Lappalainen et al., 2013*). To better understand the reasons behind this difference, we investigated three possible explanations.

### Shifts in the minor allele frequency spectrum

We observed that the minor allele frequency (MAF) spectrum of variants called in the baboon data set included proportionally more intermediate frequency variants and proportionally fewer low frequency variants than in the human data set (*Figure 2A*, inset). To investigate the degree to which this shift conferred greater power to detect eQTL in the baboons, we simulated eQTL for 10% of the genes in the study by randomly choosing a SNP near each of these genes.

We did so in two ways. First, we simulated the effect size of the eQTL, with possible effect sizes ranging from 0.25 to 2.5, in intervals of 0.25 (effect sizes are relative to a standard normal distribution). The power to detect an eQTL of a given effect size is contingent on the relative representation of different genotype classes in a population, and hence MAF (larger MAFs produce a more balanced set

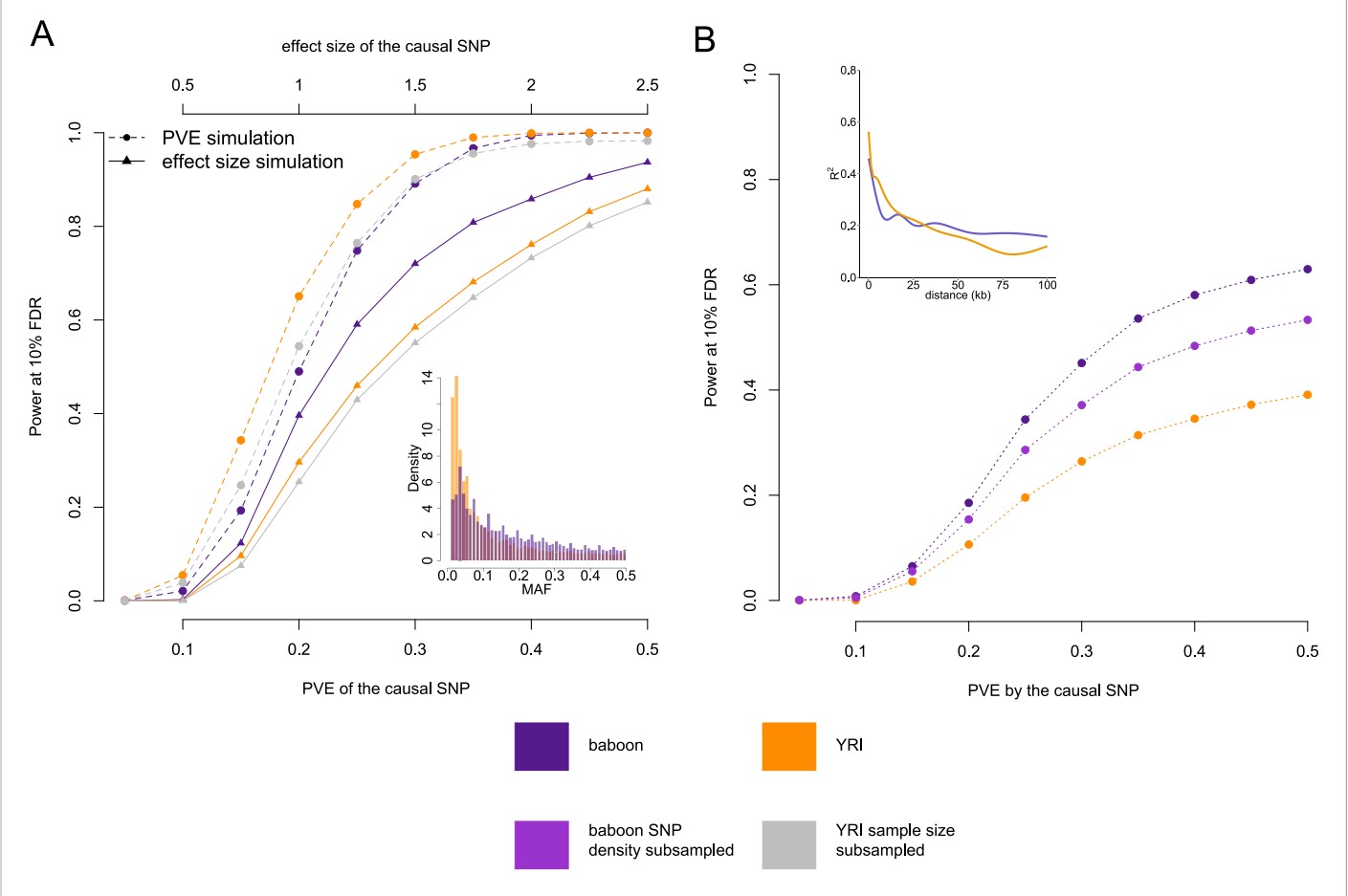

**Figure 2**. Power to detect eQTL in the Amboseli baboons compared to the HapMap YRI population. (**A**) Simulated eQTL data sets demonstrate that the baboon data set has greater power to detect eQTL (at a 10% FDR threshold) when eQTL are simulated based on effect size (solid lines and triangles) but not when eQTL are simulated based on proportion of variance in gene expression levels explained (PVE: dashed lines and circles). This result likely stems from differences in the minor allele frequency (MAF) spectrum between baboons and YRI (inset), which favors eQTL mapping in the baboons; simulations based on effect size are sensitive to MAF, but simulations based on PVE are not. (**B**) Masking the simulated eQTL SNP demonstrates that the baboon data set has greater power to detect eQTL due to both increased *cis*-regulatory SNP density and more extended LD (inset). Subsampling the SNP density in the baboon data set to the level of the YRI data set reduces the difference in power but does not remove it completely. In **B**, all results are shown for PVE-based simulations to exclude the effects of the MAF. See *Figure 4—figure supplement 1* for power simulations for masked SNPs based on effect size.

The following figure supplement is available for figure 2:

**Figure supplement 1**. Relationship between power to detect eQTL and simulated effect size, when the true eQTL is masked.

of alternative genotypes, and thus more power). Second, we simulated the proportion of variance in gene expression levels (PVE) explained by the eQTL, with possible PVE values ranging from 5% to 50%, in intervals of 5%. In this case, power to detect an eQTL does not depend on MAF because simulating the PVE directly integrates across the combined impact of effect size and MAF (a simulated high PVE eQTL with low MAF implies a large effect size variant). Thus, the impact of the MAF spectrum on the power to detect eQTL is reflected in the differences in power between the baboon data set and the YRI data set in the effect size-based vs the PVE-based simulations. In all cases, we calculated power as the proportion of genes with simulated eQTL recovered at a 10% FDR.

In PVE-based simulations, power to detect simulated eQTL was greater in the YRI data set (*Figure 2A*, dashed orange line vs dashed purple line), although this advantage disappeared when the

YRI data set was subsampled to the same size as the baboon data set (*Figure 2A*, dashed gray line). However, the baboon data set provided more power to detect eQTL than the YRI data set (whether subsampled or not) when simulations were based on effect size, where power scales with MAF (*Figure 2A*, solid lines). Based on these differences, we estimate that the power to identify an eQTL of effect size equal to the mean estimated beta in baboons (0.96), is increased in the Amboseli baboons by approximately 1.34-fold (*Figure 2A*, solid purple line vs solid orange line) as a function of differences in the MAF spectrum alone.

## Differences in genetic diversity and linkage disequilibrium

Because our RNA-seq-based approach does not identify variants outside of transcribed regions, causal SNPs were probably often not typed. To quantify the power to detect eQTL under this scenario, we again simulated eQTL among genes in the baboon and YRI data sets, but masked the causal sites. Doing so revealed much greater power to identify eQTL in baboons than in humans, across all values of simulated PVE or effect size (*Figure 2B*; *Figure 2—figure supplement 1*). One possible explanation for this observation stems from increased genetic diversity in the baboons compared to the YRI. Indeed, in baboons we tested an average of 45.4 ($\pm$57.0 s.d.) genetic variants for each gene, whereas applying the same pipeline in YRI yielded an average of 20.3 ($\pm$21.4 s.d.) testable variants per gene. An alternative explanation relates to patterns of LD, which we estimate to decay somewhat more slowly in the baboons (*Figure 2B*, inset). Higher SNP density in baboons increases the likelihood that, when a causal SNP is not typed, a nearby SNP will be available that tags it. Longer range LD suggests that a given SNP could also tag distant causal variants more effectively.

To assess the contributions of SNP density and LD, we refined our simulations by first thinning the SNP density in the baboons to match SNP density in the YRI, and again masking the simulated causal eQTL. As expected, reducing genetic diversity in the baboons reduced the power to detect genes with a true eQTL (*Figure 2B*, purple dashed line vs pink dashed line). However, it did not completely account for the difference between the human population and the baboon population, suggesting that LD patterns probably contribute to higher eQTL mapping power in baboons as well as SNP density. Specifically, for an eQTL that explains 28% of the variance in gene expression levels (the mean PVE detected in baboons for genes with significant eQTL), we estimate that SNP density and LD effects increase power by 1.21-fold (*Figure 2B*, purple dashed line vs pink dashed line) and 1.43-fold (*Figure 2B*, pink dashed line vs orange dashed line), respectively, when causal SNPs are not typed.

Together, our simulations suggest that the MAF spectrum, genetic diversity, and LD patterns increase the number of genes with detectable eQTL in baboons vs the YRI by 2.35-fold overall (1.34× from the MAF, 1.21× from SNP density effects, and 1.43× from LD effects). Further, considering that the effect size estimates in baboons tended to be larger than in the YRI (mean of 0.96 in baboons vs mean of 0.80 in YRI), the actual fold increase estimated from simulations is approximately 6-fold (*Figure 4—figure supplement 1*: ratio of purple vs orange lines at these effect sizes). This estimate is remarkably consistent with empirical results from our comparison of the real baboon and YRI data, in which we identified 6.16-fold the number of eQTL in the baboons. One possibility is that this difference arises from a history of known admixture in Amboseli between the dominant yellow baboon population and immigrant anubis baboons (*Papio anubis*: *Alberts and Altmann, 2001*; *Tung et al., 2008*). Thus, it might reflect the difference between an admixed population and an unadmixed population rather than a difference between species. However, this explanation seems unlikely because evidence for ASE does not extend further from tested genes in baboons compared to YRI (*Figure 1—figure supplement 10*), and because adding controls for local (chromosome-specific) structure when testing for eQTL still results in a large excess of eQTL detected in the baboon data set (~7× higher than in YRI: 'Materials and methods' and *Figure 1—figure supplement 11*)

## Mixed evidence for natural selection on gene expression levels

Interestingly, we found that genes harboring eQTL in baboons were also more likely to have detectable eQTL in the YRI (hypergeometric test, p = $2.39 \times 10^{-7}$). Given the sample size limitations of the data sets we considered, this overlap suggests that large effect eQTL tend to be nonrandomly concentrated in specific gene orthologues. This pattern could arise if the regulation of some genes has been selectively constrained over long periods of evolutionary time, whereas others have been

more permissible to genetic perturbation. Indeed, we found that the mean per-gene phyloP score calculated based on a 46-way primate comparison was significantly reduced (reflecting less conservation) for genes with detectable eQTL in both species, and greatest for genes in which eQTL were not detected in either case ($p < 10^{-53}$; *Figure 3A*). We obtained similar results using phyloP scores based on a 100-way vertebrate comparison ($p < 10^{-21}$; *Figure 3—figure supplement 1*).

eQTL were more likely to be identified for genes with higher genetic diversity (*Figure 1—figure supplement 8*), which may account for the relationship between phyloP score and eQTL across species: highly conserved genes are less likely to contain many variable sites. More conserved genes also tend to have slightly lower average minor allele frequencies ($p = 0.002$), which might reduce the power to detect eQTL (although the effect size is small: $r^2 = 0.001$). However, genes with eQTL in both species were also less likely to have orthologues in deeply diverged species, based on conservation in Homologene ($\beta = -0.036$, $p = 1.78 \times 10^{-8}$; *Figure 3B*). Genetic diversity within the baboons is very weakly correlated with Homologene conservation ($r^2 = 0.004$) and uncorrelated with average minor allele frequency ($p = 0.38$). Thus, sequence-level conservation scores and depth of homology across species combine to suggest that eQTL—or at least those with relatively large effect sizes—are least likely to be detected for strongly conserved loci, and most likely to be detected for lineage-specific, rapidly evolving genes. Consistent with this idea, genes involved in basic cellular metabolic processes were under-enriched among the set of genes with eQTL in both species, and enriched among the set of genes for which no eQTL were detected in either species (*Supplementary file 1B–C*). The set of genes with eQTL in either or both species, on the other hand, were enriched for loci involved in antigen processing, catalytic activity, and interaction with the extracellular environment (e.g., receptors, membrane-associated proteins).

Widespread selective constraint on gene expression levels has been suggested in previous eQTL analyses in humans, with evidence supplied by a strong negative correlation between minor allele frequency and eQTL effect size (*Battle et al., 2014*). This pattern could arise if selection acts against large genetic perturbations, such that variants of large effect would be present only at low frequencies. Consistent with this idea, plotting eQTL effect size vs MAF in the baboons results in a very strong, highly significant negative correlation ($r = -0.723$, $p < 10^{-280}$; *Figure 3C*), with no large effect eQTL detected at higher MAFs. However, such a relationship could also be a consequence of the so-called winner's curse (in which sampling variance leads to upwardly biased effect size

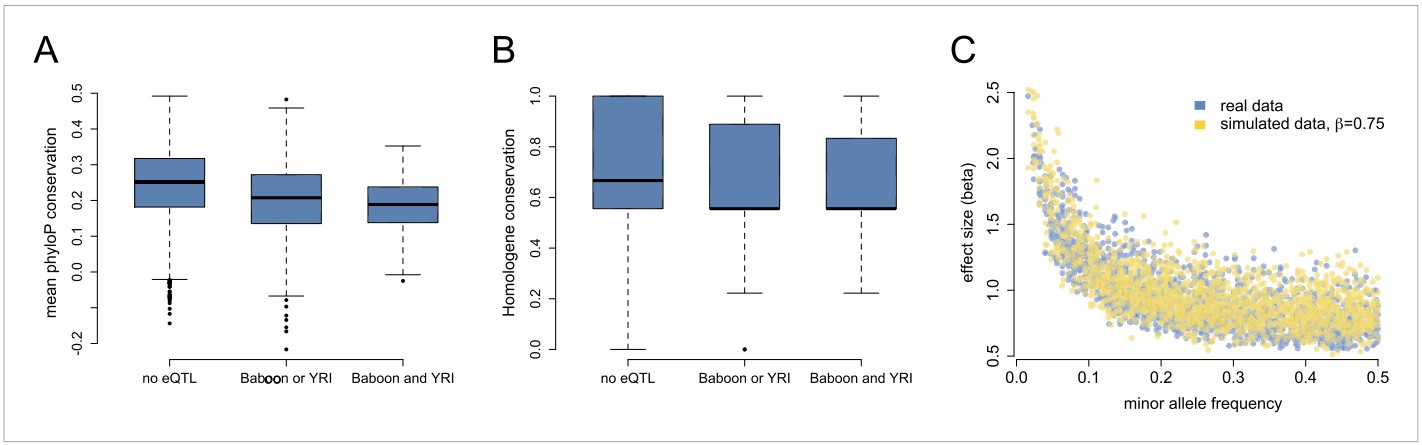

**Figure 3**. Mixed evidence for negative selection on variants affecting gene expression level. (**A**) Genes that harbor detectable eQTL in baboons, the YRI, or both are more likely to be conserved across long stretches of evolutionary time, based on mean phyloP scores in a 46-way primate genome comparison ($n = 7268$; $p < 10^{-53}$). (**B**) These genes are also more likely to be lineage-specific, based on Homologene annotations ($n = 7065$; $p = 1.78 \times 10^{-8}$). (**C**) Although we detect a strong negative correlation between eQTL effect size and eQTL minor allele frequency, in support of pervasive selection against alleles with large effects on gene expression levels, this correlation also appears when simulating constant eQTL effect sizes, suggesting winner's curse effects. See *Figure 3—figure supplement 1* for phyloP results based on a 100-way vertebrate genome comparison.

The following figure supplement is available for figure 3:

**Figure supplement 1**. Correlation between eQTL detection and mean phyloP scores based on 100-way vertebrate comparison.

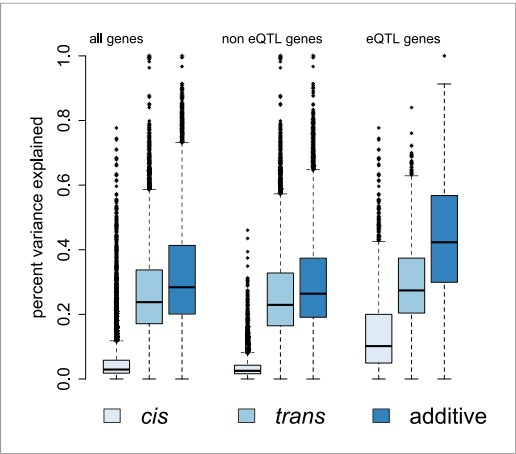

**Figure 4**. Genetic contributions to variance in gene expression levels in wild baboons. Proportion of variance in gene expression levels estimated for all genes, genes without detectable eQTL, and genes with detectable eQTL. Additive genetic effects on gene expression variation, especially *cis*-acting effects, are larger for eQTL genes than for other genes. See *Figure 4—figure supplements 1–3* for related results on percent variance explained by genetic, environmental, and demographic variables and results using an alternative set of SNPs for estimating $p_{trans}$.

The following figure supplements are available for figure 4:

**Figure supplement 1**. PVE explained by demographic and early environmental variables.

**Figure supplement 2**. Distribution of PVE explained by additive genetic variance, age, sex, and maternal social connectedness to males across all genes.

**Figure supplement 3**. Genetic contributions to variance in gene expression levels, with $p_{trans}$ based on SNPs on other chromosomes only.

estimates: *Zöllner and Pritchard, 2007*) because the degree of bias in effect size estimation is itself negatively correlated with MAF. Indeed, when we simulated sets of eQTL with constant small effect sizes ($\beta = 0.75$, close to the mean effect size detected for SNPs with MAF $\geq 0.4$), we found that the relationship between estimated effect size and MAF among detected eQTL almost perfectly recapitulated the observed negative correlation. Hence, the correlation between estimated eQTL effect size and MAF in the baboons does not provide strong support for widespread negative selection on gene expression phenotypes within species. We note, however, that our sample size of individuals is much smaller than that used for a similar analysis in humans (*Battle et al., 2014*: n = 922 individuals), and larger sample sizes should attenuate winner's curse effects.

## Genetic and environmental contributions to gene expression variation in wild baboons

Finally, we took advantage of our data set to generate the first estimates of genetic, demographic (age and sex), and environmental contributions to gene expression variation in wild nonhuman primates (*Supplementary file 1D*). While our limited sample size leads to high variance around estimates for any individual gene, the median estimates across genes should be unbiased (*Zhou et al., 2013*), so we concentrated on these overarching patterns. We focused specifically on three social environmental variables of known importance in this population, all of which have been extensively investigated as models for human social environments. These were: (i) early life social status, which predicts growth and maturation rates (*Altmann and Alberts, 2005*; *Charpentier et al., 2008*); (ii) maternal social connectedness to other females, which predicts both adult lifespan and the survival of a female's infants (*Silk et al., 2003*, *2009*, *2010*; *Archie et al., 2014*); and (iii) maternal social connectedness to males, based on recent evidence that heterosexual relationships have strong effects on survival as well (*Archie et al., 2014*).

Overall, we found that genetic effects on gene expression levels tended to be far more pervasive than demographic and environmental effects. Specifically, the median additive genetic PVE was 28.4%, similar to, or slightly greater than, estimates from human populations (*Monks et al., 2004*; *McRae et al., 2007*; *Emilsson et al., 2008*; *Price et al., 2011*; *Wright et al., 2014*). We applied a Bayesian sparse linear mixed model (BSLMM: *Zhou et al., 2013*) to further partition this additive genetic PVE into two components: a component attributable to *cis*-SNPs (here, all SNPs within 200 kb of a gene) and a component attributable to *trans*-SNPs (all other sites in the genome). Again similar to humans (*Price et al., 2011*; *Wright et al., 2014*), we found that more of the additive genetic PVE is explained by the *trans* component (median PVE = 23.8%) than the *cis* component (median PVE = 2.9%) (*Figure 4*). Unsurprisingly, we estimated a larger *cis*-acting component for genes in which functional *cis*-regulatory variation

was detected in our previous analysis (median PVE = 10.2% among eQTL genes and median PVE = 5.0% among ASE genes).

In contrast to the substantial genetic effects we detected, the median PVE explained by age and sex were 1.89% and 0.82%, respectively (*Figure 4—figure supplements 1–2*). The distribution of PVE explained by age was significantly greater than expected by chance (Kolmogorov–Smirnov test on binned PVEs, in comparison to permuted data: p < 10$^{-11}$), whereas that explained by sex was not (p = 0.100); large sex effects tended to be constrained to a small set of genes on the X chromosome (*Figure 4—figure supplement 1*). Of the early environmental variables we investigated, only maternal social connectedness to males explained more variance in gene expression levels than expected by chance (p = 4.19 × 10$^{-3}$), with a median PVE of 1.9%. Notably, while social connectedness to males (i.e., heterosexual bonds) and social connectedness to females (i.e., same-sex bonds) are both known predictors of longevity in the Amboseli baboons, previous analyses suggest that their effects are largely independent (*Archie et al., 2014*). Our result extends this observation to the early life effects of maternal social connectedness on variance in gene expression levels.

Taken together, our data suggest that while almost all genes are influenced by genetic variation, the effects of demographic and environmental parameters are generally modest for any single aspect of the environment. However, in at least some cases, we find evidence that early environmental effects on gene expression levels appear to persist across the life course, as has previously been demonstrated in laboratory settings and in response to severe early adversity in humans (e.g., *Weaver et al., 2004*; *Miller et al., 2009*; *Cole et al., 2012*).

## Discussion

Much of what we know about genetic contributions to variation in gene expression levels in primates (and vertebrates more generally) come from the extensive body of research on humans. However, increasing evidence indicates that humans are demographically unusual: compared to other primates, humans exhibit low levels of neutral genetic diversity and a low long-term effective population size (*Chen and Li, 2001*; *Hernandez et al., 2007*; *Perry et al., 2012*). Further, humans are distinguished from other primates by recent explosive population growth (*Keinan and Clark, 2012*; *Tennessen et al., 2012*). While late Pleistocene population expansion has been suggested for some nonhuman primates, including chimpanzees and Chinese-origin rhesus macaques (*Hernandez et al., 2007*; *Wegmann and Excoffier, 2010*), none have undergone the extreme levels of population increase that characterized humans. Indeed, evidence from microsatellite data suggests that the long-term effective population size of baboons actually may have contracted during this period (*Storz et al., 2002*).

These differences are not simply of historical interest, but also important for understanding the genetic architecture of traits measured in the present day. Differences in demographic history not only affect overall levels of genetic variation and the minor allele frequency spectrum, but also the mean effect size of sites that contribute to phenotypic variation (*Lohmueller, 2014*). Interestingly, demographic history does not impact overall trait heritability (*Lohmueller, 2014*; *Simons et al., 2014*), perhaps explaining why we estimated mean additive genetic PVEs for gene expression levels in baboons that are similar to those estimated for humans. However, demographic history can influence the power to detect individual genetic contributions to phenotypic variation. Large-scale population expansion of the type that occurred in human history appears to reduce power to identify genotype-phenotype correlations for fitness-related traits (*Lohmueller, 2014*). This observation may account, in part, for our ability to identify many more functional regulatory variants in the baboons than we expected based on previous studies in humans.

However, while our analysis extends previous observations that large effect eQTL are nonrandomly distributed, we found mixed evidence for widespread negative selection on gene expression levels. Specifically, within the baboons alone, we found that the negative relationship between eQTL effect size and minor allele frequency was explicable based on winner's curse effects alone. Thus, increased power to identify functional regulatory variants in the baboons is probably not due to pervasive associations between gene expression levels and fitness. In contrast, stronger evidence for selection on gene expression patterns stems from our cross species comparisons. In particular, we observed that genes with eQTL in baboons significantly overlapped with genes with eQTL in humans, and that these genes as a class also tended to be less constrained at the sequence-level (consistent with observations for analyses of *cis*-eQTL in humans alone: *Popadin et al., 2014*). This result suggests that

genes vary in their tolerance of functional regulatory genetic variation, and, intriguingly, that gene-specific robustness to genetic perturbation may be a conserved property across species.

Because no comparable data are yet available for other large mammal populations, including for other baboons, it is unclear whether our results are typical or instead a consequence of the Amboseli population's own unique history. In particular, the population has experienced recent admixture between yellow baboons, the dominant taxon, and closely related anubis baboons (*P. anubis*) (*Alberts and Altmann, 2001*; *Tung et al., 2008*). Admixture, which appears to be relatively common in natural populations (*Mallet, 2005*), can have important consequences for genetic diversity and LD patterns. While it appears to have had a modest impact on the relative ability to map gene expression phenotypes in baboons vs the YRI data set, comparison to a non-admixed baboon population could help resolve this question further. More generally, our results encourage further investigation of the relationship between demography and trait genetic architecture in other populations, as has been suggested for humans (*Lohmueller, 2014*) but could also be profitably extended to nonhuman model systems. Such comparisons would provide an empirical basis for testing predicted relationships between demographic history and the power to identify genotype-phenotype associations. From an applied perspective, they could also help identify animal models that favor more highly powered association mapping studies, a strategy that has already been heavily exploited in domestic dogs (*Karlsson et al., 2007*; *Karlsson and Lindblad-Toh, 2008*) and suggested for rhesus macaques (*Hernandez et al., 2007*). While the same sites will probably rarely be associated with the same traits across species, this strategy could help identify molecular mechanisms that are conserved across humans and animal models (e.g., *Lamason et al., 2005*). Comparisons that use matched sample types will be particularly informative: our study compared eQTL detection in whole blood (from the baboons) with eQTL from lymphoblastoid cell lines (YRI), which exhibit highly correlated, but not identical, patterns of overall gene expression (Spearman's rho = 0.645, $p < 1 \times 10^{-16}$)—these differences could affect rates of eQTL detection as well.

Finally, our data—the first profile of genome-wide gene expression levels in a wild primate population—serve as a useful proof of principle of the ability to concurrently generate genome-wide gene expression phenotype and genotype data, and to relate them to each other using eQTL and ASE approaches. Intensively studied natural primate populations—some of which have been studied continuously for 30 or more years—have emerged as important phenotypic models for human behavior, health, and aging. The approach we used here provides a way to leverage these models for complementary genetic studies as well, especially if eQTL prove to be strongly enriched for sites associated with other traits, as in humans (*Nicolae et al., 2010*). Although preliminary, our results highlight the increasing feasibility of integrating functional genomic data with phenotypic data on known individuals in the wild. For example, our data set revealed a number of genes in which variation in gene expression levels could be mapped to an identifiable eQTL, validated using an ASE approach, and also linked to early life environmental variation. Such cases suggest the potential for future investigations of the molecular basis of persistent environmental effects, including whether genetic and environmental effects act additively or interact.

## Materials and methods

### Study subjects and blood sample collection

Study subjects were 63 individually recognized adult members (26 females and 37 males) of the Amboseli baboon population. All study subjects were recognized on sight by observers based on unique physical characteristics. To obtain blood samples for RNA-seq analysis, each baboon was anesthetized with a Telazol-loaded dart using a handheld blowpipe. Study subjects were darted opportunistically between 2009 and 2011, avoiding females with dependent infants and pregnant females beyond the first trimester of pregnancy (female reproductive status is closely monitored in this population, and conception dates can be estimated with a high degree of accuracy). Following anesthetization, animals were quickly transferred to a processing site distant from the rest of the group. Blood samples for RNA-seq analysis were collected by drawing 2.5 ml of whole blood into PaxGene Vacutainer tubes (Qiagen, Valencia, CA), which contain a lysis buffer that stabilizes RNA for downstream use. Following sample collection, study subjects were allowed to regain consciousness in a covered holding cage until fully recovered from the effects of the anesthetic. They were then released within view of their

social group; all subjects promptly rejoined their respective groups upon release, without incident.

Blood samples were stored at approximately 20°C overnight at the field site. Samples were then shipped to Nairobi the next day for storage at −20°C until transport to the United States and subsequent RNA extraction.

## Gene expression profiling using RNA-seq

For each RNA sample (one per individual), we constructed an RNA-seq library suitable for measuring whole genome gene expression using Dynal bead poly-A mRNA purification and a standard Illumina RNA-seq prep protocol. Each library was randomly assigned to one lane of an Illumina Genome Analyzer II instrument and sequenced to a mean depth of 30 million 76-base pair reads (±4.5 million reads s.d., *Supplementary file 1A*). The resulting reads were mapped to the baboon genome (*Panu2.0*) using the efficient short-read aligner *bwa 0.5.9* (*Li and Durbin, 2009*), with a seed length of 25 bases, a maximum edit distance of two mismatches in the seed, a read trimming quality score threshold of 20, and the default maximum edit distance (4% after trimming). To recover reads that spanned putative exon–exon junctions, and therefore could not be mapped directly to the genome, we used the program *jfinder* on reads that did not initially map (*Pickrell et al., 2010b*). Finally, we filtered the resulting mapped reads data for low quality reads (quality score <10) and for reads that did not map to a unique position in the genome. To assign reads to genes, we used the RefSeq exon annotations for *Panu* 2.0 (*ref_Panu_2.0_top_level.gff3*, downloaded September 6, 2012). We considered the total read counts for each gene and individual as the sum of the number of reads for that individual that overlapped the union of all exon base pairs assigned to a given gene. In downstream analyses, we considered only highly expressed genes that had non-zero counts in more than 10% individuals, and that had mean read counts greater than or equal to 10 (excluding the gene for beta-globin).

We then performed quantile normalization across samples followed by quantile normalization for each gene individually, resulting in estimates of gene expression levels for each gene that were distributed following a standard normal distribution. This procedure effectively removed GC bias in gene expression level estimates (*Figure 1—figure supplement 2*). For eQTL mapping, ASE analysis, and PVE estimation for sex and age we used all 63 individuals. For PVE estimation for maternal rank and social connectedness, missing data meant that we conducted our analysis on n = 52 and n = 47 individuals, respectively.

## Variant identification and genotype calls

To identify genetic variants in the baboon data set, we used the Genome Analysis Toolkit (v. 1.2.6; *McKenna et al., 2010*; *DePristo et al., 2011*). Because no validated reference set of known genetic variants are available for baboon, we performed an iterative bootstrapping procedure for base quality score recalibration. Specifically, we performed an initial round of base quality score recalibration and identified a set of variants using GATK's UnifiedGenotyper and VariantFiltration walker. From this call set, we constructed a set of high confidence variants with quality score ≥100 that passed all filters for variant confidence (variants failed if QD < 2.0), mapping quality (variants failed if MQ < 35.0), strand bias (variants failed if FS > 60.0), haplotype score (variants failed if HaplotypeScore >13.0), mapping quality (variants failed if MQRankSum < −12.5) and read position bias (variants failed if ReadPosRankSum < −8.0). We used this high confidence set as the set of 'known sites' in a second round of base quality score recalibration, repeating this procedure until the number of variants identified in consecutive rounds of recalibration stabilized. In the final call set, we removed all sites (i) that were monomorphic in the Amboseli samples; (ii) for which genotype data were missing for more than 12 individuals (19%) in the data set; (iii) that deviated from Hardy–Weinberg equilibrium; and (iv) that failed the above quality control filters. We further filtered the data set to contain only sites with a minimum quality score of 100 that were located within 200 kb of a gene of interest, and that were sequenced at a mean coverage ≥5× across all samples. We validated our quality control and filtering steps by performing the same procedure on an RNA-seq data set from the HapMap Yoruba population (see below). These steps resulted in a set of 64,432 single nucleotide polymorphisms carried forward into downstream analysis (30,938 for the YRI). For eQTL mapping analysis, missing genotypes in this final set were imputed using BEAGLE (*Browning and Browning, 2009*).

To estimate genome-wide LD, we followed the approach of *Eberle et al. (2006)*, which uses allele frequency-matched SNPs to calculate pair-wise LD. Specifically, we selected SNPs with MAFs greater than 10% and divided them into four subgroups (MAF between 10%–20%; MAF between 20%–30%; MAF between 30%–40%; and MAF between 40%–50%). We then calculated pair-wise $r^2$ for all SNP pairs within 100 kb in each subgroup using VCFtools (*Danecek et al., 2011*) and combined values from all four subgroups.

## Estimating accuracy of SNP genotypes using human RNA-seq data

To assess the accuracy of the RNA-seq-based genotyping calls we performed in the baboons, we investigated a similarly sized data set of RNA-seq reads from a human population (*Pickrell et al., 2010a*). Because this data set focused on samples from the HapMap consortium (n = 69 members of the Yoruba population from Ibadan, Nigeria), we were able to compare genotypes called using the RNA-seq pipeline to independently collected genotype data from HapMap Phase 3 (r27) (*International HapMap Consortium, 2010*). To do so, we focused on 9919 variants that were genotyped in both data sets. We then calculated the correlation between genotypes called in the RNA-seq-based pipeline and genotypes from HapMap, for each individual (*Figure 1—figure supplement 5A*). We also found that low accuracy was correlated with the level of apparent homozygosity in the genotype data (*Figure 1—figure supplement 5B*). In the baboon data, we had no individuals with unusually low homozygosity, but six individuals with unusually high homozygosity (>80% of genotype calls). These outliers were missing a median of 10.6% of data in the unimputed genotype data set, whereas all other individuals were missing a median of 0.6% data. However, removing these six individuals from our analysis resulted in very similar results as using the full data set: 87.6% of eQTL genes (n = 1566) identified when using all individuals were also identified with this subset.

Importantly, the available data from humans also support accurate variant discovery. Of the 30,938 sites that we identified from the RNA-seq data and that passed all of our filters, only 3.1% (967) did not have an assigned rsID in dbSNP release 138. These sites were likely enriched for false positives, as the transition/transversion ratio for this set was 1.42, vs 2.80 for the set of 30,938 sites as a whole.

## eQTL mapping

To identify *cis*-acting eQTLs in the baboon data set, we used the linear mixed model approach implemented in the program GEMMA (*Zhou and Stephens, 2012*). This model provides a computationally efficient method for eQTL mapping while explicitly accounting for genetic non-independence within the sample; in our case, some individuals in the data set are related (although overall relatedness was low: the median kinship coefficient across all pairs was 0.015; mean = 0.024 ± 0.033 s.d.).

For each gene, we considered all variants within 200 kb of the gene as candidate eQTLs. For each variant, we fitted the following linear mixed model:

$$y = \mu + x\beta + u + \varepsilon,$$

$$u \sim MVN(0, \ \sigma_u^2 K),$$

$$\varepsilon \sim MVN(0, \ \sigma_e^2 I),$$

and tested the null hypothesis $H_0$: $\beta = 0$ vs the alternative $H_1$: $\beta \neq 0$. Here, y is the *n* by 1 vector of gene expression levels for the *n* individuals in the sample. Gene expression values were first corrected for hidden factors that could act as sources of global structure (e.g., batch effects or ancestry- or environment-related *trans* effects) by regressing out the first 10 principal components of the gene expression data. Consistent with previous results (e.g., *Pickrell et al., 2010a*), this procedure greatly improves our ability to detect eQTL (*Figure 1—figure supplement 12*). In the model, $\mu$ is the intercept; x is the *n* by 1 vector of genotypes for the variant of interest; and $\beta$ is the variant's effect size. The *n* by 1 vector of u is a random effects term to control for individual relatedness and other sources of population structure, where the *n* by *n* matrix K = XX$^T$/p provides estimates of pairwise relatedness derived from the complete 63 × 64,432 genotype data set X. Residual errors are represented by $\varepsilon$, an n by 1 vector, and MVN denotes the multivariate normal distribution.

We took the variant with the best evidence (i.e., lowest p-value) for association with gene expression levels for each gene, and then calculated corrected gene-wise q-values (with a 10% false discovery rate threshold) via comparison to the same values obtained from permuted data (similar to *Barreiro et al., 2012*; *Pickrell et al., 2010a*).

## Possible confounds associated with eQTL mapping using RNA-seq data

We evaluated the potential for eQTL mapping based on RNA-seq data to introduce four possible confounds.

First, for genes with large effect *cis*-eQTLs, reads from heterozygotes at eQTL-linked sites might be biased towards the allele associated with higher gene expression levels. If so, heterozygotes might be mistakenly genotyped as homozygotes for the high expressing allele, resulting in an underrepresentation of heterozygous genotypes relative to neutral expectations. To control for this possibility, we eliminated sites that violated Hardy-Weinberg expectations (n = 2386) from our analyses. We note, however, that this scenario would not introduce false positives. Instead, it would lead to more conservative detection of additive eQTL effects, with the direction of an estimated eQTL effect still consistent with the true effect.

Second, SNP calling might be biased towards the reference allele. If so, more reads would be required to support a genotype call of homozygote alternate than a genotype call of homozygote reference. This bias would result in higher apparent expression levels for alternate allele homozygotes and lower expression levels for reference allele homozygotes, which could create false positive eQTLs. However, we observe no evidence for this scenario in our data set. For all tested SNPs (n = 64,432) and for eQTL SNPs only (n = 1693), alternate allele homozygotes tend to have slightly lower coverage than reference allele homozygotes, and heterozygotes tend to have the highest coverage (because more reads are required to support inference of heterozygosity) (*Figure 1—figure supplement 13*). Thus, coverage and genotype do not covary additively, and this potential confound is unlikely to produce false positive eQTLs.

Third, read mapping might be biased towards the reference allele, such that reads carrying the alternate allele are less likely to map because they contain more mismatches to the reference genome. This possibility is consistent with our observation that alternate allele homozygotes tend to have slightly less coverage than reference allele homozygotes (*Figure 1—figure supplement 13*). While this difference in coverage is significant (Kolmogorov-Smirnov test: $p < 2.2 \times 10^{-16}$ for all SNPs; $p = 3.9 \times 10^{-5}$ for eQTL SNPs), the magnitude of the effect itself is modest (*Figure 1—figure supplement 13*), probably because we allowed reads to map with up to three mismatches: Wittkopp and colleagues have shown that reference allele mapping bias is largely obviated by allowing reads to map with more mismatches (*Stevenson et al., 2013*). Further, systematic calling of false positive eQTLs due to biased read mapping would predict a bias towards negative effect sizes (i.e., eQTL effects suggesting that the alternate allele is associated with lower expression levels). Our data are not consistent with such a pattern: 47% of eQTL betas are negative, whereas 53% are positive. Reference allele mapping biases are, however, more likely to affect ASE analysis, producing a pattern of greater expression in the reference allele. Indeed, we do observe a bias towards negative betas in the ASE analysis (67.2% of n = 510 genes), although the overall magnitude and direction of ASE data agree well with eQTL evidence.

Fourth, lower mean coverage in homozygotes of either type relative to heterozygotes could induce false positive eQTLs in which the major allele was associated with lower gene expression levels. To test this possibility, we recoded eQTL effects to reflect the effect of the major allele instead of the effect of the alternate allele (i.e., a genotype of 0 = homozygous minor and a genotype of 2 = homozygous major). We observed a modest excess of eQTL for which the major allele was associated with lower gene expression levels (56%, binomial test $p = 1.15 \times 10^{-7}$). This bias did not differ depending on whether the major allele was the reference allele or the alternate allele (Fisher's Exact Test, p = 0.28), supporting minimal read mapping biases in our data. Instead, it appears to be primarily driven by SNPs with low minor allele frequencies (proportion of negative betas for the lowest quartile of MAFs = 62.8%, $p = 7.49 \times 10^{-8}$; highest quartile of MAFs = 48.6%, p = 0.602). At these sites, eQTL inference relies primarily on two genotype classes (the major allele homozygotes and heterozygotes) rather than three genotype classes. Because heterozygotes tend to have slightly higher coverage than homozygotes of both classes, spurious

relationships between genotype and gene expression levels are much less likely to be observed when both types of homozygotes are well represented (i.e., MAFs are larger).

Along with the high genotype accuracy rates estimated from the Yoruba data, our analyses thus indicate that the set of eQTL we identified are largely robust to RNA-seq-specific confounds. The eQTL identified in YRI in the conventional pipeline vs the RNA-seq pipeline offer a further source of comparison. We find that eQTL identified through the RNA-seq pipeline tend to be associated with more highly expressed genes (providing greater power to call genotypes: Wilcoxon test p = $6.53^{-9}$), but otherwise do not differ in sequence conservation (phyloP scores: p = 0.707; Homologene scores: p = 0.603) or in estimated effect size (p = 0.137) (*Figure 1—figure supplement 9*). Further, effect size magnitude is highly correlated across pipelines when eQTL are discovered in both pipelines (r = 0.874, p < $10^{-57}$). When eQTL were discovered only in the RNA-seq pipeline (n = 104), they tended to be high on the ranked list of eQTL evidence in the conventional pipeline as well (median rank of 1395, where the top 700 were significant and 10,615 genes were tested), suggesting that many of them did not pass the threshold for eQTL detection in that analysis. Thus, the most salient source of error stems from low MAF sites, which are also the cases most vulnerable to sampling error and winner's curse effects more generally (*Figure 3*)—a problem that is not confined to RNA-seq-based eQTL mapping. Taken together, these analyses argue that, as a general rule, eQTL associated with lower MAF SNPs should be treated with increased caution.

## ASE detection

To identify ASE, we focused on SNPs within gene exons with Phred-scaled quality scores greater than 10. We further required that these sites have more than five reads in more than two individuals and more than 300 total reads across all heterozygous individuals. This threshold is based on the observation that the power to detect ASE is dependent on sequencing read coverage at heterozygous sites (*Fontanillas et al., 2010*). Indeed, in our data set, power to detect ASE appeared to scale primarily with total read coverage rather than number of heterozygous individuals. Sites with more reads tended to have more heterozygotes (r = 0.266, p < $10^{-100}$); however, when sites were partitioned by total read depth (in deciles), sites with significant ASE were not more likely to harbor more heterozygotes in any decile (Wilcoxon test comparing number of heterozygotes in significant sites vs background; *Figure 1—figure supplement 14*).

After these filtering steps, we retained 8154 SNPs associated with 2280 genes for ASE analysis. For ASE analysis, we did not take into account possible recombination between exonic SNPs and the (unknown) *cis*-regulatory variants whose effects they capture, as we did not have detailed data on recombination rates across the baboon genome. However, recombination between exonic SNPs and the true causal regulatory SNPs would decrease our power to detect ASE.

For each variant, we considered a beta-binomial distribution (following *Pickrell et al., 2010a*) to model the number of reads from the (+) haplotype (denoted as $x_i^+$) or the number of reads from the (−) haplotype (denoted as $x_i^-$), conditional on the number of total reads (denoted as $y_i = x_i^+ + x_i^-$), for each individual i, or

$$x_i^+ | y_i \sim binomial(y_i, \theta),$$

$$\theta \sim beta(\alpha, \beta).$$

We tested the null hypothesis $H_0$: $\alpha = \beta$ vs the alternative $H_1$: $\alpha \neq \beta$ using a likelihood ratio test. For both the null model and the alternative model, beta distribution parameters ($\alpha$ and $\beta$) were estimated via a maximum likelihood approach, using the R function *optim*. Again, we took the variant with the lowest p-value for each gene, and then calculated corrected gene-wise q-values (using a 10% false discovery rate threshold) via comparison to the same values obtained from an empirical null distribution. To construct the empirical null distribution, we performed the same analysis after substituting the $x_i^+$ value for each variant of interest, for each heterozygous individual, with a randomly selected $x_i^+$ value from a heterozygous site elsewhere in the genome (contingent on that site having the same number of total reads, $y_i$).

## Power simulations

To assess the relative power of eQTL mapping in baboons vs the YRI data set, we randomly selected 10% of the genes in each data set to harbor eQTL. For each of these simulated eQTL genes, we then

randomly chose a SNP among all the *cis*-SNPs tested (i.e., all variable sites that passed quality control filters and fell within 200 kb of a gene of interest) and assigned it as a causal eQTL. The impact of the eQTL was simulated using either effect size, in which we simulated a constant effect size between 0.25 and 2.5 (in intervals of 0.25) or PVE, in which we chose an effect size that explained a specific proportion of variance in gene expression levels (from 5% to 50%, in intervals of 5%). We then simulated gene expression levels by adding the effect of the simulated *cis*-eQTL SNP to residual errors drawn from a standard normal distribution. To calculate the FDR, we also simulated a set of genes with no eQTL. For each combination of effect sizes and population (baboon or YRI), and for each simulation scenario (e.g., with the causal SNP masked or unmasked, with SNP density thinned in the baboons, or using PVE vs a constant effect size), we performed 10 replicates. For each replicate, we calculated the power to detect eQTL as the proportion of simulated eQTL genes recovered at a 10% empirical FDR.

## Testing the contribution of admixture to eQTL detection

To investigate whether admixture might drive our power to detect eQTL in the baboon data set, we performed three analyses.

First, we asked whether evidence for ASE remained similar across longer distances (i.e., between sites separated by more base pairs) in the baboons vs in the YRI. Such a pattern might be expected if long-distance, admixture-driven LD explained our other observations. However, the pattern of ASE similarity (the magnitude of the difference between ASE estimates) by distance between sites was highly congruent between the YRI and baboon data sets (*Figure 1—figure supplement 10*).

Second, we investigated whether adding a control for local structure (i.e., population structure in *cis* to a gene of interest, and based only on variants located on the same chromosome) asymmetrically reduced evidence for eQTL in the baboon data set relative to the YRI data set. To do so, we regressed out the top two PCs for variants on the same chromosome as the gene of interest from the gene expression data prior to fitting mixed effects models. We found that this approach modestly reduced the number of eQTL discoveries in the baboon data set (n = 1583 from n = 1787, an 11.4% difference). However, this number was still 5.4× larger than the number of eQTL detectable in the YRI, and when we applied the same local structure control to the YRI data, a comparable drop in the number of discoverable eQTL also occurred (n = 216 from n = 290, resulting in a ~7× fold increase in eQTL in baboons vs YRI).

Third, we compared the spatial distribution of eQTL in baboon between the models with and without local structure controls. We reasoned that if admixture drove most of the signal in the data set, controlling for local structure should shift the location of discovered eQTL closer to the gene of interest, where the strongest *cis* effects are generally identified. However, the locations of eQTL were very similar under both models (Kolmogorov-Smirnov test, p = 0.577; *Figure 1—figure supplement 11*).

## Evidence for patterns consistent with natural selection on gene expression levels

We investigated the relationship between conservation level and the presence of detectable eQTL in the Amboseli baboons or the YRI using phyloP conservation scores (*Pollard et al., 2010*) and Homologene conservation of orthology across species. For the former, we extracted the per-site phyloP score from the 46-way primate comparison or 100-way vertebrate comparison on the UCSC Genome Browser for each base contained within the annotated exons (including untranslated regions) used for mapping RNA-seq reads in the YRI. We then calculated the average phyloP score across all exons associated with a given gene. We obtained Homologene scores from the CANDID database (*Hutz et al., 2008*). In both cases, we used linear models to test for a relationship between conservation level and three categories of genes: those with no detectable eQTL in either the baboons or YRI; those with a detectable eQTL in one of the two species; and those with a detectable eQTL in both species.

To investigate whether the correlation between minor allele frequency and eQTL effect size could be a result of winner's curse effects, we extracted the results from our simulations in which the causal variant was masked and the true effect size was fixed at a small value (beta = 0.75). We then calculated the correlation between the estimated effect size ($\beta$) from these simulations against minor allele frequency, for detected eQTL only.

## Estimation of genetic contributions to gene expression

We used the Bayesian sparse linear mixed model (BSLMM) approach implemented in the GEMMA software package (*Zhou and Stephens, 2012*) to estimate the genetic contribution to gene expression variation. Specifically, for each gene, we fit the following model:

$$y = \mu + x_{cis}\beta_{cis} + x_{trans}\beta_{trans} + \varepsilon,$$

$$\beta_{cis,i} \sim \pi N\left(0, \ \sigma_a^2\right) + (1-\pi)\delta_0,$$

$$\beta_{trans,i} \sim N\left(0, \ \sigma_b^2\right),$$

where y is the *n* by 1 vector of gene expression levels for *n* individuals; μ is the intercept; $x_{cis}$ is an n by $p_{cis}$ matrix of genotypes for $p_{cis}$ *cis*-SNPs and $\beta_{cis}$ are the corresponding effect sizes; $x_{trans}$ is an n by $p_{trans}$ matrix of genotypes for $p_{trans}$ *trans*-SNPs and $\beta_{trans}$ are the corresponding effect sizes; and ε is an n by 1 vector of i.i.d. residual errors. We used different priors for *cis*-acting effects and *trans*-acting effects to capture different properties for the two components. Specifically, the spike-slab prior on the *cis* effects $\beta_{cis}$ captures our prior belief that only a small proportion of local SNPs has *cis* effects and these effects are relatively large. The normal prior on the *trans* effects captures our prior knowledge that *trans*-acting SNPs tend to be relatively difficult to find and have relatively small effects. In addition, because $p_{cis}$ is small and $p_{trans}$ approximately equals p, the number of total SNPs, we used p instead of $p_{trans}$ to facilitate computation (i.e., $p_{trans}$ was based on all genotyped sites used in our analyses, n = 64,432). Results are qualitatively similar if $p_{trans}$ is calculated based on sites that must act in *trans* (i.e., sites located on a different chromosome than the chromosome containing the gene of interest: *Figure 4—figure supplement 3*). We used Markov chain Monte Carlo (MCMC) to fit the model with 1000 burn-in and 10,000 sampling steps. We obtained posterior samples of $\beta_{cis}$ and $\beta_{trans}$ to calculate the PVE attributed by each of the two components, as well as the total additive genetic PVE contributed by both components.

To calculate PVE values for demographic and environmental predictors, we again used the linear mixed model approach implemented in GEMMA to control for additive genetic effects. Sex was known from direct observation of the study subjects. Ages were known to within a few days' error for 52 of the 63 individuals in the data set; six animals had birth dates estimated to be accurate within 1 year, four animals had birth dates estimated to be accurate within 2 years, and one had a birth date estimated to be less accurate than 2 years. Early social status was measured using the proportional dominance rank of the individual's mother, at the time of that individual's conception. Dominance ranks are assigned monthly using *ad libitum* observations of dyadic agonistic (aggressive or competitive) encounters within social groups (*Hausfater, 1974*; *Alberts et al., 2003*). Maternal social connectedness values were defined as the social connectedness of the individual's mother, in the year of that female's life during which the focal individual was born. Social connectedness is calculated on a yearly basis as the frequency with which a female was involved in affiliative interactions, relative to the median for all females in the population at the same time and controlling for observer effort (see *Runcie et al., 2013*; *Archie et al., 2014*). Social connectedness is measured for females, but can focus on either female–female relationships (SCI-F) or a female's relationship with adult males (SCI-M), which have independent effects on longevity in this population (*Archie et al., 2014*). For SCI-F, affiliative interactions included both grooming interactions and close spatial proximity to other females. For SCI-M, only grooming interactions were used.

For each gene, we fit the following model:

$$y = \mu + x\beta + u + \varepsilon,$$

$$u \sim MVN\left(0, \ \sigma_u^2 K\right),$$

$$\varepsilon \sim MVN\left(0, \ \sigma_e^2 I\right),$$

where x is the *n* by 1 vector of values for the demographic or environmental predictor of interest and *β* is its coefficient. The *n* by 1 vector of u is a random effects term with K = XX$^T$/p controlling for additive genetic effects. We calculated the PVE estimate as var(x*β*)/var(y), where var denotes the sample variance.

## Acknowledgements

We thank the Kenya Wildlife Services, Institute of Primate Research, National Museums of Kenya, National Council for Science and Technology, members of the Amboseli-Longido pastoralist communities, Tortilis Camp, and Ker & Downey Safaris for their assistance in Kenya. We also thank J Altmann for general support and access to the Amboseli data set and samples; RS Mututua, S Sayialel, JK Warutere, M Akinyi, T Wango, and V Oudu for invaluable assistance with sample collection; K Michelini for assistance with RNA-seq data generation; S Mukherjee for advice on statistical analysis; and LB Barreiro, S Montgomery, the associate editor, and one anonymous reviewer for comments on an earlier draft of the manuscript. Finally, we thank the Baylor College of Medicine Human Genome Sequencing Center for access to the current version of the baboon genome assembly (*Panu 2.0*).

## Additional information

### Funding

| Funder | Grant reference number | Author |
| --- | --- | --- |
| National Institute on Aging | AG034513 | Susan C Alberts |
| National Science Foundation (NSF) | IOS 0919200 | Susan C Alberts |
| National Institute of General Medical Sciences (NIGMS) | GM077959 | Yoav Gilad |
| National Human Genome Research Institute (NHGRI) | HG006123 | Yoav Gilad |
| University of Chicago | | Jenny Tung |
| National Institute on Aging | AG031719 | Susan C Alberts |
| National Science Foundation (NSF) | DEB 0846286 | Susan C Alberts |

The funders had no role in study design, data collection and interpretation, or the decision to submit the work for publication.

### Author contributions

JT, Conception and design, Acquisition of data, Analysis and interpretation of data, Drafting or revising the article; XZ, YG, Conception and design, Analysis and interpretation of data, Drafting or revising the article; SCA, Acquisition of data, Drafting or revising the article, Contributed unpublished essential data or reagents; MS, Analysis and interpretation of data, Drafting or revising the article

### Ethics

Animal experimentation: Samples and data used in this study were collected from wild baboons living in the Amboseli ecosystem of southern Kenya. All behavioral, environmental, and demographic data are gathered as part of noninvasive observational monitoring of known individuals within the study population. This research is conducted under the authority of the Kenya Wildlife Service (KWS), the Kenyan governmental body that oversees wildlife (permit number NCST/5/002/R/777 to SCA and NCST/RCD/12B/012/57 to JT). As the animals are members of a wild population, KWS requires that we do not interfere with injuries to study subjects inflicted by predators, conspecifics, or through other naturally occurring events (e.g., falling out of trees). To collect blood samples, we perform temporary immobilizations using the anesthetic Telazol, delivered via a handheld blowgun. Permission to perform this procedure was granted through KWS, and was performed under the supervision of a KWS-approved Kenyan veterinarian, who monitored anesthetized animals for hypothermia, hyperthermia, and trauma (no such events occurred during our sample collection efforts). Observational and sample collection protocols were approved though IACUC committees at Duke University (current protocol is A028-12-02 to SCA and JT) and the University of Chicago (ACUP 72080 to YG).

# Additional files

## Supplementary file

• Supplementary file 1. Supplementary tables. (**A**) Read mapping summary. (**B**) Gene Ontology analysis for genes with no eQTL in baboon or YRI. (**C**) Gene Ontology analysis for genes with eQTL in either or both baboon and YRI. (**D**) Demographic and environmental data.

## Major datasets

The following data set was generated:

| Author(s) | Year | Dataset title | Dataset ID and/or URL | Database, license, and accessibility information |
|---|---|---|---|---|
| Tung J, Zhou X, Alberts SC, Stephens M, Gilad Y | 2015 | The Genetic Architecture of Gene Expression Levels in Wild Baboons | GSE63788 | Publically available at the NCBI Gene Expression Omnibus (http://www.ncbi.nlm.nih.gov/geo/). |

The following previously published data set was used:

| Author(s) | Year | Dataset title | Dataset ID and/or URL | Database, license, and accessibility information |
|---|---|---|---|---|
| Pickrell JK, Marioni JC, Pai AA, Degner JF, Engelhardt BE, Nkadori E, Veyrieras JB, Stephens M, Gilad Y, Pritchard JK | 2010 | Understanding mechanisms underlying human gene expression variation with RNA sequencing | GSE19480 | Publically available at the NCBI Gene Expression Omnibus (http://www.ncbi.nlm.nih.gov/geo/). |

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
