## [Decision Letter]

Thank you for sending your work entitled “The Genetic Architecture of Gene
Expression Levels in Wild Baboons” for consideration at *eLife*.
Your article has been favorably evaluated by Aviv Regev (Senior editor), a Reviewing
editor, and two reviewers, one of whom, Stephen Montgomery, has agreed reveal his
identity.

The Reviewing editor and the reviewers discussed their comments before we reached this
decision, and the Reviewing editor has assembled the following comments to help you
prepare a revised submission.

Please find below a summary of the comments of the reviewers that we request that you
address in your revised manuscript:

Figures 1 and 3 should not be main figures.
The pipeline should be moved to the supplement. Also the ASE and eQTL effect sizes is a
general technical issue and does not relate specifically to the study of the genetics of
gene expression in baboons. The text around these sections is long and generally
distracting.

There is a comment regarding the lack of an association between effect size and minor
allele frequency compared to [12]. However, Battle et al. had 922 individuals compared to 66 individuals
here (among several technical differences). We are not sure that these studies can be
easily compared to provide a definitive statement regarding negative selection in
baboons here. A caveat regarding study differences might suffice.

The authors bring up the possibility of admixture in the baboon population. This is
slightly concerning as it could increase the number of hets and has the potential to
increase the length of haplotypes (both observations of the study). It further could
explain many of the observations of the study. Three possible suggestions to address
this are to test whether: (i) one sees an excess of trans-associations on the same
chromosome compared to across chromosomes for cis-eQTL in baboons versus YRI, (ii) apply
a surrogate variable or hidden factor correction to eQTL analysis on a single chromosome
or (iii), our reviewers’ favorite, test if ASE is correlated over a longer
distance (more independent genes) than in humans—in this case, since there is no
biological basis for why the locations of causal variants should be further away from
the genes they regulate between baboons and humans, the decay of the correlation of ASE
measured in proximal genes should be similar. We realize the authors have a model to
control for individual relatedness and population structure, but this is derived from
the entire genetic data set and does not address local patterns of admixture.

The gene expression data was quantile normalized. Why was a hidden factor correction not
applied? Typically, these types of corrections dramatically improve eQTL discovery. Our
concern is if there is some structure to the data that is both present and correlated in
genotype and gene expression space, the number discoveries will be artificially
inflated.

For ASE analyses: (i) the authors assume no recombination, this is not stated, (ii) how
is beta in theta∼beta (alpha, beta) estimated, and (iii) detection of ASE
correlates with expression level (Figure 3),
this is not a surprise, but given their model we are concerned whether this estimate is
more extreme, because ASE has different variances in effect size when it is estimated
from a few individuals for very highly expressed genes (2 het individuals with 150 reads
= 300 total) compared to lots of estimates from intermediately expressed genes
(10 het individuals with 30 reads = 300 total). For robustness, the authors
should show whether detection of ASE in their study is independent of the number of
input individuals once a testable site has been selected using their criterion.

The authors should discuss the possibility that the negative correlation between
conservation and probability of eQTL in a gene in baboons at least may be driven by the
technical issue that only coding SNPs were tested and therefore conserved genes will
tend to have low MAF and therefore low power.

The authors indicate a large component of expression variability is in
*trans*. Is *trans* defined as on other chromosomes? In
particular, the authors should clarify what goes into the p_trans_ matrix.

---

## [Author Response]

Figures 1 and 3
*should not be main figures. The pipeline should be moved to the supplement. Also
the ASE and eQTL effect sizes is a general technical issue and does not relate
specifically to the study of the genetics of gene expression in baboons. The text
around these sections is long and generally distracting*.

We have removed the original Figure 1 from the
manuscript, as it provided an overview version of the more detailed pipelines provided
in the original Figure 1—figure supplement 1, Figure 1—figure supplement 2, Figure 1—figure supplement 3, Figure 1—figure supplement 4, Figure 1—figure supplement 5 and Figure 1—figure supplement 6. These supplements are now attached to
the original Figure 2 (now Figure 1), along with other figures that were previously included
as Figure 1’s supplements. The plots in
former Figure 3 have also been changed to Figure 1 supplements and removed from the main text.
In addition, we have streamlined our discussion of the power to detect ASE versus eQTL
in the main text, largely removing this entire section.

*There is a comment regarding the lack of an association between effect size and
minor allele frequency compared to*
[12]*.
However, Battle et al. had 922 individuals compared to 66 individuals here (among
several technical differences). We are not sure that these studies can be easily
compared to provide a definitive statement regarding negative selection in baboons
here. A caveat regarding study differences might suffice*.

We agree and have added such a caveat to this section of the Results.

*The authors bring up the possibility of admixture in the baboon population. This
is slightly concerning as it could increase the number of hets and has the potential
to increase the length of haplotypes (both observations of the study). It further
could explain many of the observations of the study. Three possible suggestions to
address this are to test whether: (i) one sees an excess of trans-associations on the
same chromosome compared to across chromosomes for cis-eQTL in baboons versus YRI,
(ii) apply a surrogate variable or hidden factor correction to eQTL analysis on a
single chromosome or (iii), our reviewers’ favorite, test if ASE is correlated
over a longer distance (more independent genes) than in humans—in this case,
since there is no biological basis for why the locations of causal variants should be
further away from the genes they regulate between baboons and humans, the decay of
the correlation of ASE measured in proximal genes should be similar. We realize the
authors have a model to control for individual relatedness and population structure,
but this is derived from the entire genetic data set and does not address local
patterns of admixture*.

Thanks for these helpful suggestions. We agree that the evolutionary history of this
population could have influenced our findings. In the revised manuscript, we investigate
this possibility in substantially greater detail (eleventh paragraph in the Results
section and under the heading “Testing the contribution of admixture to eQTL
detection”, in a new Materials and methods section). Overall, we found little
evidence that admixture drove our results. These conclusions are based on three sets of
findings:

1) Distance between sites tested for ASE predicts the magnitude of the difference
between ASE estimates, as expected; however, this relationship does not differ between
baboons and YRI (new Figure 1—figure supplement 10). This analysis, motivated by the reviewers’ third
suggestion, suggests that ASE is not correlated over a longer distance in baboons than
in humans (note that we could not directly test how correlations between ASE estimates
decay with distance because our ASE estimates are site-specific, not
individual-specific).

2) Controlling for local structure in addition to global ancestry (suggestion 2)
modestly reduces the number of eQTL discoveries in the baboon data set, but not more so
than the same procedure in YRI. When controlling for local structure using the top two
principal components for variants on the same chromosome, the number of detectable eQTL
drops from 1787 to 1583. However, this number is still 5.4-fold larger than the number
detected in the YRI. A drop also occurs when running a parallel model in YRI (from 290
to 216), so that the number of eQTL detected in baboons is ∼7-fold larger when
controls for local structure are used in both. Thus, more extensive local structure does
not appear to explain the increased power in baboons.

3) In support of this idea, the spatial distribution of baboon eQTLs relative to genes
is almost identical between the models with and without local structure controls (new
Figure 1—figure supplement 11). If
admixture drove most of the signal in the data set, we would expect to observe greater
enrichment of eQTL within or near genes when adding controls for local structure; we do
not.

*The gene expression data was quantile normalized. Why was a hidden factor
correction not applied? Typically, these types of corrections dramatically improve
eQTL discovery. Our concern is if there is some structure to the data that is both
present and correlated in genotype and gene expression space, the number discoveries
will be artificially inflated*.

This comment reflects our mistake in writing the original version of our Methods. We
indeed corrected for hidden factors by regressing out the first 10 principal components
of the overall gene expression data. As the reviewers note, this process greatly
improved our ability to detect eQTL and minimized the possibility that the eQTL we
detected reflect global structure in gene expression space (we did the same for the YRI
data, so our methods remained comparable). We explain our procedure in the revised
Methods (tenth paragraph) and have also added a figure supplement (new Figure 1—figure supplement 12) showing the
relationship between the number of PCs we removed and eQTL detection.

*For ASE analyses: (i) the authors assume no recombination, this is not stated,
(ii) how is beta in theta∼beta (alpha, beta) estimated, and (iii) detection of
ASE correlates with expression level (*Figure 3*), this is not a surprise, but given
their model we are concerned whether this estimate is more extreme, because ASE has
different variances in effect size when it is estimated from a few individuals for
very highly expressed genes (2 het individuals with 150 reads = 300 total)
compared to lots of estimates from intermediately expressed genes (10 het individuals
with 30 reads = 300 total). For robustness, the authors should show whether
detection of ASE in their study is independent of the number of input individuals
once a testable site has been selected using their criterion*.

With regards to (i), we did not make any explicit assumptions about recombination in the
ASE analysis. Recombination between the exonic SNPs we used to test for ASE and true
causal regulatory SNPs would decrease our power to detect ASE; we make this point clear
in the revision (in the subsection “ASE detection” of the Materials and
methods). For (ii), we used a maximum likelihood approach to estimate both the alpha and
beta parameters in the beta component of the beta-binomial model; we have clarified this
point in the same subsection of the Materials and methods of the revised manuscript. To
address point (iii), we have now tested whether ASE is more likely to be detected for
sites with more heterozygous individuals, conditional on total read depth (overall,
number of hets and total read depth are correlated: r = 0.266, p <
10^-100^). We find no evidence for such an effect across the ten deciles of
total read depth values we tested (please see the aforementioned subsection and Figure 1—figure supplement 14).

*The authors should discuss the possibility that the negative correlation between
conservation and probability of eQTL in a gene in baboons at least may be driven by
the technical issue that only coding SNPs were tested and therefore conserved genes
will tend to have low MAF and therefore low power*.

Thanks for pointing out this possible interpretation. We have now calculated the
correlation between levels of conservation and average minor allele frequency for each
gene, for SNPs used in our analysis. The correlation between a gene’s average
phyloP score (from the 46-way primate comparison) and the average MAF of all SNPs tested
in association with that gene is nominally significant (p = 0.002) but explains a
very small fraction of the variance (r = -0.037). Thus, while more conserved
genes do tend to have lower average MAFs (among SNPs tested), this relationship is weak,
probably because we filter out sites with very low MAFs and because many sites occur in
non-protein coding regions of the transcript or in the transcribed regions of other
genes. Further, we observe no significant correlation between a gene’s Homologene
score and average minor allele frequency among SNPs tested (p = 0.38). These
analyses suggest that the relationship between conservation and eQTL discovery is
probably not driven by a relationship between conservation and MAF (at least within the
data set we ultimately analyzed). We have discussed this point in the revised manuscript
(under the heading “Mixed evidence for nature selectin of gene expression
levels”, in the Results section). We also cite a recent paper by Popadin et al.
(third paragraph of the Discussion) that also suggests that fewer
*cis*-eQTL are found in older genes, similar to our findings for the
Homologene analysis.

*The authors indicate a large component of expression variability is in*
trans*. Is* trans *defined as on other chromosomes? In
particular, the authors should clarify what goes into the
p*_*trans*_
*matrix*.

Because the number of sites that are included in p_trans_ is almost equal to p
(the total number of sites in the genome), we calculated the p_trans_ matrix
based on all SNPs typed. However, in the revision we also have run p_trans_
based on all other chromosomes except the chromosome containing the focal gene. These
results are almost identical to the results obtained when using all SNPs; we include
them as a new supplemental figure (please see the subsection “Estimation of
genetic contributions to gene expression”, in the Materials and methods, and new
Figure 4—figure supplement 3).